



# Basal thermal regime affects the biogeochemistry of subglacial systems

Ashley Dubnick[1], Martin Sharp[1], Brad Danielson[1,2], Alireza Saidi-Mehrabad[3], Joel Barker[4]

[1] Department of Earth and Atmospheric Science, University of Alberta, Edmonton AB, T6G 2E3, Canada
[2] Fiera Biological Consulting, Suite 301, 10359-82 Ave, Edmonton AB, T6E 1Z9
[3] Department of Biological Sciences, University of Alberta, Edmonton AB, T6G 2E3, Canada
[4] School of Earth Sciences, The Ohio State University, Marion 43302, USA

*Correspondence to*: Ashley Dubnick (adubnick@ualberta.ca)

## Abstract

Ice formed in the subglacial environment can contain some of the highest concentrations of solutes, nutrients, and microbes found in glacier systems. Upon glacial melt, these materials are released to downstream freshwater and marine ecosystems and glacier forefields. Despite the potential ecological importance of basal ice, our understanding of its biogeochemical characteristics, and their spatial and temporal variability, remains limited. We hypothesize that the basal thermal regime of glaciers is a dominant control on subglacial biogeochemistry because it influences the

degree to which glaciers mobilize material from the underlying substrate and controls the nature and extent of biogeochemical activity that occurs at glacier beds. Here, we characterize the solutes, nutrients, and microbes found in the basal regions of a cold-based glacier and three polythermal glaciers and compare them to those found in overlying glacier ice. Compared to its parent glacier ice, basal ice from polythermal glaciers was consistently enriched in major ions, dissolved organic matter (including a specific fraction of humic-like fluorescent material), and

microbes, and occasionally enriched in dissolved phosphorus and reduced nitrogen ($NH_4^+$) and in a second dissolved component of humic-like fluorescent material. In contrast, the biogeochemistry of basal ice from the cold-based glacier was remarkably similar to that of its parent glacier ice. Although basal ice from the cold-based glacier may have acquired some inorganic and organic nutrients from the underlying substrate, it did not appear to contain significant amounts of either solutes or microbes derived from the glacier bed. These findings suggest that a glacier's

basal thermal regime can play an important role in determining the mix of solutes, nutrients, and microbes that are acquired from subglacial substrates and/or produced *in situ*.

## 1 Introduction

Glaciers form by the compression and metamorphism of snow and slowly deform and flow under their own weight. While a considerable portion of a glacier's ice is of meteoric origin, receiving chemical and biological inputs primarily from the

atmosphere, ice can also form at or near the bed as glaciers erode bedrock and/or subglacial sediments and entrain both





particulate and dissolved materials. This basal ice typically acquires solutes (often dominated by $Ca^{2+}$, $Mg^{2+}$, $HCO_3^-$ and $SO_4^{2-}$), via reactions that involve carbonate and sulphide minerals, which are trace components in most types of bedrock (Holland, 1978). It may also incorporate organic matter and nutrients (e.g. phosphorus, silica, potassium) and microbes from the underlying substrate (Montross et al., 2014; Sharp et al., 1999). Both basal ice and subglacial water may also host populations

of microbes that mediate redox reactions (e.g. Sharp et al., 1999), play an active role in bedrock weathering (e.g. Tranter et al., 2002), and produce and/or consume ecologically important nutrients such as sediment-bound phosphorus (e.g. Hodson, 2007), iron (e.g. Statham et al., 2008), sulphate (e.g. Bottrell and Tranter, 2002), silica (e.g. Tranter et al., 2002), nitrogen (e.g. Boyd et al., 2011), and organic carbon (e.g. Wadham et al., 2012).

In glaciers where surface-derived meltwater drains through the subglacial environment and comes into contact with basal ice

and subglacial sediments, it will mix meltwater produced at the bed. These processes can dramatically alter the geochemistry (Tranter et al., 2002), nutrient content (Hawkings et al., 2014; Wadham et al., 2016) and microbial community composition (Dubnick et al., 2017) of meltwater during its transit from the glacier surface to downstream proglacial environments. During glacial retreat, melting of basal ice releases the sediment, solutes, nutrients and microbes. These materials contribute to the nutrient dynamics of glacier forefields (Kazemi et al., 2016; Mindl et al., 2007; Sattin et al., 2010) and form the basis of the

soils from which many postglacial landscapes evolve (Kastovská et al., 2005).

Despite the relatively high concentration and/or unique composition of solutes, nutrients, and microbes often found in subglacial systems, and their potential to impact glacier forefields and downstream ecosystems, our understanding of subglacial biogeochemical processes and products, and their spatio-temporal variability, remains limited. Basal temperature can vary considerably within and between glaciers and it controls rates of ice deformation and basal sliding, the amount of water near

the bed and thus the extent to which glaciers mobilize material from the underlying substrate, and the extent and characteristics of *in situ* biogeochemical activity. We therefore hypothesize that the basal thermal regime plays an important role in defining the physical and biogeochemical characteristics and variability of basal ice. Since warm ice deforms more easily than cold ice, and subglacial water promotes basal sliding (Iken, 1981; Iken and Bindschadler, 1986), we expect basal ice that forms and persists in fast-flowing glaciers to experience relatively 'warm' conditions and have distinct biogeochemistries from basal ice

that forms and persists in the relatively 'cold' conditions of slow-flowing glaciers. To evaluate how basal thermal regime affects the biogeochemical materials that glaciers mobilize from the substrate or produce/cycle within subglacial environments, we explore the solutes, nutrients and microbes found in the basal regions of three fast-flowing, polythermal outlet glaciers and the slow-flowing Western Margin of the Devon Ice Cap (DIC, Devon Island, Nunavut, Canada).

## 2 Methods

### 2.1 Study Site

The Devon Ice Cap (DIC) covers an area of approximately 14,400 $km^2$ (Burgess and Sharp, 2004) and has been shrinking since 2005 (Sharp et al., 2011). We collected and characterized basal ice from three fast-flowing outlet glaciers (Sverdrup



Glacier, Belcher Glacier, and East 7 Glacier) and the slow-flowing Western Margin of the ice cap (Fig. 1). All three fast-flowing glaciers have surface velocities > 20 m a$^{-1}$ and a temporally varying component of flow which peaks in summer and

has been attributed to seasonal variation in the rate of basal motion (Burgess et al., 2005; Van Wychen et al., 2017) (Fig. 1). The occurrence of time-varying basal motion in these glaciers suggests that geothermal and frictional heat keep ice near the glacier bed at temperatures that are at or near the pressure-melting point over a considerable portion of their beds (Burgess et al., 2005; Cuffey and Paterson, 2010). Further, field observations of hydrologically active moulins on the surfaces of these glaciers suggests that surface-derived meltwater likely reaches at least some areas of their beds during the melt season. The

presence of large open subglacial meltwater channels beneath the lateral margins of Sverdrup and Belcher glaciers suggests channelized subglacial water drainage. In May, prior to the initiation of surface melt, air temperatures at distances of >500 m into these subglacial channels were near 0 ℃. We therefore assume that the basal ice in these fast-flowing glacier systems likely formed and persisted under relatively 'warm' subglacial conditions, and we refer to it as 'warm' basal ice. Ice at the Western Margin of the ice cap is relatively slow-flowing (with surface velocities generally <10 m a$^{-1}$ (Burgess et al., 2005))

and is not constrained laterally by bedrock topography. We infer that glacier flow in this region occurs exclusively by internal deformation with ice frozen to the bed (Burgess et al., 2005). Therefore, unlike the 'warm' basal ice that forms and persists at the beds of the faster-flowing glaciers, the basal ice at the Western Margin is likely below the pressure melting point and is referred to here as 'cold' basal ice. Although located up to ~100 km apart, the three fast-flowing glaciers explored in this study are all underlain by metasedimentary rocks and gneiss, while ice at the Western Margin is largely underlain by sandstone,

dolomite and limestone bedrock (Harrison et al., 2016) (Fig. 1).

## 2.2 Field Sampling

We sampled 'warm' basal ice from the three fast-flowing glaciers, and 'cold' basal ice from the Western Margin of the ice cap, as well as overlying meteoric glacier ice from each (Fig. 1). We identified basal ice near the glacier bed as an ice facies with high debris content and an anisotropic structure that incorporated features such as discontinuous layers, lenses and pods

of varying size (Hubbard and Sharp, 1989; Knight, 1997). In contrast, glacier ice is typically white, bubby and horizontally stratified. Ice samples were collected from marginal ice cliff faces, ice rubble at the base of cliff faces, and the walls of subglacial meltwater channels. All ice samples were collected using sterile (furnaced at 500 ℃ for 8 hrs) carbide chisels that were ethanol-bathed and flame-sterilized in the field before each use. One chisel was used for debris-poor samples and another was used for debris-rich samples. At least 10 cm of material was removed from exposed surfaces in the field before samples

were collected. Samples were stored in 5L Whirl-pak bags (Nasco, Fort Atkinson, USA) and kept frozen (~ -20℃) until analysis.

## 2.3 Sample Processing

Prior to analysis, samples were removed from the freezer and melted at 4 ℃ in Whirl-pak bags. For 16S rRNA gene sequencing, a glass filter tower and 0.2 µm Pall Supor® polysulfone 47 mm sterile filter papers were used to filter samples. Filter papers





were collected in duplicate and stored in sterile petri dishes at -80ºC until further processing.  For analyses of soluble reactive

phosphorus (SRP), $NO_3^- + NO_2$, and $NH_4^+$ water analyses, a sterile syringe (60 ml) with a 0.45 µm cellulose acetate luer-lok

filter was used to fill two sterile 15 ml centrifuge tubes. One 120 ml Nalgene® bottle was also filled (with no headspace) for

major ion analyses and stored at 4 ºC for ~2 weeks until analysis. For dissolved organic carbon (DOC) quantification and

characterization, a 0.7 µm GF/F luer-lok filter was used to fill two sterile 45 ml universal glass vials (leaving headspace) and

a piece of foil was placed beneath the cap for closure before freezing.

All filtration equipment was rinsed 3 times with sample, and a minimum of 5 ml of sample was passed through each filter

paper before the sample was filtered for analysis. Glassware was acid-washed (10% HCl for >48 hrs), and both glassware and

foil were combusted (450ºC for 8 hrs) prior to use. All storage bottles, lids, and foil caps were rinsed 3 times with filtrate

before a sample was collected for analysis.

**2.4 Analytical Methods**

*Nutrient concentrations ($SRP/TDP/NH_4^+/NO_3^-+NO_2^-/TDN/SiO_2$):* Determinations of soluble reactive phosphorus (SRP) and

total dissolved phosphorus (TDP), ammonium ($NH_4^+$), nitrate ($NO_3^-$) + nitrite ($NO_2^-$), total dissolved nitrogen (TDN) and

reactive silica ($SiO_2$) were made with a Lachat QuickChem QC 8500 FIA Automated Ion Analyzer (Lachat Instruments,

Loveland, CO, USA) using methods outlined by Rice *et al.* (2012) and O'Dell (1993) for $NO_3^- + NO_2^-$. Detection limits were

TDP = 1.8 ppb, SRP = 0.9 ppb, $NH_4^+$ = 3 ppb, $NO_3^- + NO_2^-$ = 2 ppb, TDN = 7 ppb, and $SiO_2$ = 0.02 ppm.

*DOC concentrations:* dissolved organic carbon was quantified using a Shimadzu TOC-5000A Total Organic Carbon Analyzer

(Shimadzu, Japan) equipped with a high-sensitivity platinum catalyst using US EPA method # 415.1. The detection limit for

DOC was 0.2 mg $L^{-1}$.

*DOM characteristics:* We used three-dimensional Excitation Emission Matrices (EEMs) derived from total fluorescence scans

to broadly characterize dissolved organic matter (DOM) into humic-like and protein-like fractions and to correlate specific

fluorophores with those previously identified in the literature. DOM fluorescence was measured in ratio mode (S/R) using an

Agilent G1321B fluorescence detector (Agilent Technologies, Santa Clara, USA) and methods outlined by Cuss and Gueguen

(2015). Prior to each analysis, the system was rinsed 3 times with deionized water and 3 times with sample at room temperature.

EEMs were produced by measuring the fluorescence intensity every 1 nm at excitation wavelengths from 220-450 nm and

every 5 nm at emission wavelengths from 280-545 nm.

*Major Ions ($Cl^-$, $SO_4^{2-}$, $Na^+$, $K^+$, $Mg^{2+}$, $Ca^{2+}$):* Anions were quantified using a Dionex DX-600 Ion Chromatograph (Dionex,

USA) and methods outlined by US EPA method # 300.1. Cations were measured using Inductively Coupled Plasma - Optical

Emission Spectroscopy (ICP-OES; Thermo Scientific iCAP 6300, Cambridge, UK) and US EPA method #200.7. Detection

limits were $Cl^-$ = 0.85 µeq $L^{-1}$, $SO_4^{2-}$ = 0.83 µeq $L^{-1}$, $Na^+$=0.87 µeq $L^{-1}$, $K^+$=0.26 µeq $L^{-1}$, $Mg^{2+}$=0.82 µeq $L^{-1}$, and $Ca^{2+}$= 0.5

µeq $L^{-1}$.





*Sediment concentration:* 50 ml of unfiltered, melted ice was placed in a pre-weighed 50 ml dish and dried at 50 ℃. The dish was then reweighed and the sediment in 50 ml was calculated as the change in mass from before to after the sample was added/dried.

*16S rRNA gene sequencing:* DNA was extracted from filter papers using MO BIO's PowerSoil ® DNA Isolation kit following
the manufacturer's protocol, but with several modifications to maximize the efficiency of the extraction, including: 1) at Step 14, solution C4 was added for a total of 4 ml instead of 1,200µl and vortexing was for 20 seconds instead of 5 seconds, 2) prior to step 15, the samples were incubated for 30 minutes, and 3) at step 20, the DNA was eluted in 50µl of solution C6 instead of 100µl. Primers 515F and 806R were used to amplify V4 region of the 16S rRNA gene. The Illumina libraries were denatured and diluted following Illumina guidelines. An 8 pM library containing 7% PhiX each was sequenced on a MiSeq instrument
(Department of Biology, University of Waterloo) using a 2 x 250 cycle Reagent Kit v2.

### 2.5 Data Processing and Statistical Analyses

*Geochemistry and inorganic nutrients:* The concentration of $HCO_3^-$ (µeq $L^{-1}$) was calculated as the charge deficit between the sum of cations (µeq $L^{-1}$) and the sum of anions (µeq $L^{-1}$). To summarize the geochemical composition of the solute load in each sample, we (i) calculated the fractional contribution of each major ion to the total solute load by dividing the concentration
of each major ion (µeq $L^{-1}$) by either the sum of cations or the sum of the anions in the respective sample, (ii) normalized the fractional contributions of each ion species by their respective mean and variance (Iwamori et al., 2017) and (iii) conducted a Principal Components Analysis (PCA) in Matlab R2018a using these data. To summarize the nutrient composition of each sample and evaluate the concentrations of TDP and TDN relative to the total solute load, the concentrations of TDP (µg P $L^{-1}$) and TDN (µg N $L^{-1}$) were divided by the solute concentration (µeq $L^{-1}$) of the corresponding sample. Spearman rank-order
correlation coefficients ($r_s$) were used to evaluate the significance of dependency between geochemical/nutrient variables.

*DOM Characterization:* Parallel Factor Analysis (PARAFAC) was used to decompose the complex EEMs into discrete components using the drEEM toolbox in Matlab2018a and methods developed by Murphy et al. (2013). Corrections were applied for instrument spectral bias and for inner filter effects, and Raman scatter was normalized to daily Raman scans (Murphy et al., 2013). The scatter region for each EEM was excised and smoothed and EEMs were normalized to unit variance.
PARAFAC was completed using non-negativity constraints and the EEM normalization was reversed after modelling. Although the modelled components cannot be identified as specific organic compounds, they were characterized using the OpenFluor database (Murphy et al., 2014) and comparisons with previous literature. To summarize the DOM composition of each sample, the fluorescent intensity of each component was normalized to its mean and variance across the dataset and a PCA was completed.

*Microbial Assemblage:* Paired-end reads were assembled using PANDAseq (Masella et al., 2012) and analysed using Quantitative Insights Into Microbial Ecology (QIIME, (Caporaso et al., 2010)), managed by the automated exploration of microbial diversity v. 1.5 (AXIOME, (Lynch et al., 2013)). Sequences were clustered with UPARSE (Edgar, 2013) and compressed into unique operational taxonomical units (OTUs) with 97% similarity, and classified by The Ribosomal Database





Project (RDP) (Wang et al., 2007) with a confidence threshold of 0.8. Rarefaction analysis was used to sub-sample the processed dataset to lower than the smallest library for subsequent analyses. The microbial assemblages in each sample were summarized by completing non-metric multidimensional scaling (NMDS) of Bray-Curtis distance measures, statistical significance between groups was determined using multi response permutation procedure (MRPP) and multiple linear regressions to fit environmental vectors onto the NMDS (using the Vegan toolbox in R).

## 3 Results

### 3.1 Major Ion Chemistry

Glacier ice and 'cold' basal ice had relatively low solute concentrations ($\bar{x}$ = 15.6 µeq L$^{-1}$ and 22 µeq L$^{-1}$, respectively) that were dominated by atmospherically-derived solutes, including Cl$^-$, SO$_4^{2-}$, and Na$^+$ (Table 1). 'Warm' basal ice samples contained significantly higher concentrations of solutes ($\bar{x}$ = 241 µeq L$^{-1}$), including common rock-derived solutes such as K$^+$, Ca$^{2+}$, Mg$^{2+}$, and HCO$_3^-$, than did glacier ice (T-test, $p<0.05$) (Table 1; Fig. 3). Therefore, while the composition and
concentration of solutes in 'cold' basal ice were very similar to those in glacier ice, it is likely that distinct solute sources exist in relatively 'warm' subglacial systems (Fig. 2).

### 3.2 Nutrients (N and P)

While 'warm' basal ice did not contain significantly more dissolved inorganic nutrients than glacier ice, including nitrogen (TDN, NH$_4^+$ or NO$_3^-$) and phosphorus (SRP, TDP) (T-Test, $p>0.05$), it contained, on average, less NO$_3^-$ (T-test, $p<0.05$) and
had higher inter-sample variability in the concentrations of NO$_3^-$, and NH$_4^+$ and TDP (F-Test, $p<0.05$; Table 1, Fig. 3). Therefore, while the subglacial system of 'warm' based glaciers may function as a sink of NO$_3^-$, the sources (and potentially also sinks) of other inorganic nutrients may be spatially heterogeneous. 'Cold' basal ice samples had significantly higher concentrations of NH$_4^+$, TDN, SRP, TDP than samples of parent glacier ice (T-test, $p<0.05$; Table 1), suggesting the existence of a relatively consistent subglacial source for these nutrients in this 'cold' subglacial system.

### 3.3 Dissolved Organic Matter

A 5 component PARAFAC model explained 98.6% of the variability in the spectrofluorescence dataset. Two of these components were similar to protein-like fluorescence and three were similar to humic-like fluorescence as described in other studies (Table 2). DOC concentrations in 'warm' basal ice were not significantly different from those in glacier ice (T-test, $p<0.05$) but DOC concentrations in 'warm' basal ice were positively correlated with the tyrosine-like C1 fluorescence (r$_s$
=0.61, $p=0.01$, n=9). 'Warm' basal ice also contained significantly more humic-like C3 fluorescence (T-test, $p<0.05$), and significantly more variable humic-like C3 and C5 fluorescence (*F*-test, $p<0.05$) than did glacier ice (Table 1). Thus, a relatively consistent fraction of the DOC derived from these subglacial systems was probably in the form of proteinaceous and humic organic matter.



'Cold' basal ice had significantly higher DOC concentrations ($\bar{x}$ = 0.40 ppm) than glacier ice ($\bar{x}$ = 0.15 ppm) (T-test, $p<0.05$;
Table 1), suggesting that ice can acquire DOC in 'cold' basal environments. Relative to glacier ice, 'cold' basal ice also
contained significantly higher and more variable fluorescence of humic-like DOM components C3 and C5 (T-test and F-Test,
$p<0.05$) and exhibited significantly more variable fluorescence of protein-like DOM component C2 than did glacier ice (*F*-
test, $p<0.05$; Table 1). Since C3 and C5 humic-like DOM is typically associated with terrestrial soils and vegetation (Table 2),
this DOM may have been acquired during past glacial advances over soils and sediments.

**3.4 Microbial Assemblages**

Microbial assemblages in 'warm' basal ice and glacier ice formed two distinct groups (Fig. 2). 76% of the OTUs observed in
'warm' basal ice were absent from glacier ice (Fig. 4), suggesting that a large portion of the microbial assemblage in 'warm'
basal ice was sourced from the subglacial environment. Microbial assemblages in 'warm' basal ice were also highly variable
- less than half a percent of the OTUs found in 'warm' basal ice were present in all 'warm' basal ice samples. Geographic
location influenced the structure of microbial assemblages in 'warm' basal ice since 'warm' basal ice samples from a given
glacier had more OTUs in common with other basal ice samples from the same glacier (42% of their OTUs and 26% of their
assemblages) than they did with basal ice samples from other 'warm' based glaciers (32% of their OTUs and 15% of their
assemblages) (T-test, $p<0.001$). Furthermore, although multiple linear regressions did not indicate any significant correlations
between the major ion concentrations, major ion composition, nutrient concentrations or fluorescence index and the microbial
assemblage structure of 'warm' basal ice samples, sample location was significantly correlated with the structure of microbial
assemblages in 'warm' basal ice samples ($p=0.01$).

The microbial assemblages in 'cold' basal ice were broadly similar to those in glacier ice (Fig. 2). 'Cold' basal ice shared most
(73%) OTUs with glacier ice (Fig. 4). Of the shared OTUs between 'cold' basal ice and glacier ice, many (37%) were absent
from the 'warm' basal ice samples. Thus, the microbial assemblages in 'cold' basal ice remained remarkably similar to those
in its parent material, despite the potential for interactions with the substrate.

**4 Discussion**

**4.1 Basal ice formation**

Glacier ice originates as snow in the accumulation zones of glaciers/ice caps. This ice is of meteoric origin and
receives chemical and biological inputs primarily from the atmosphere, experiences consistently sub-freezing temperatures
and is likely to host limited *in situ* biogeochemical activity in the englacial system. Surface melt routed through the subglacial
system and/or subglacially-produced meltwater formed by geothermal and frictional heat sources may refreeze to form basal
ice beneath temperate and polythermal glaciers. The interactions between ice/water and the overridden substrate can mobilize
sediment, solutes, microbes, and nutrients and incorporate them into the base of the glacier during the formation of 'warm'
basal ice. Relatively warm temperatures (ie: near the pressure melting point) beneath the glacier may also promote



biogeochemical activity by increasing both the availability of liquid water and the metabolic rates of micro-organisms, and rates of chemical weathering.

The subglacial conditions in cold-based glaciers that are frozen to their beds differ considerably from those in temperate and polythermal glaciers because temperatures are below the pressure melting point. The modes of formation of 'cold' basal ice can vary between glaciers and are generally poorly understood. Thus, interpretations of the environments in, and processes by, which such ice is formed are often ambiguous, making our understanding of the biogeochemistry of 'cold' basal ice even more limited. The formation of 'cold' basal ice is often described by the 'apron entrainment model' that invokes the production of basal ice by the overriding and reworking of apron material (snow, ice blocks, refrozen melt water and debris) along an advancing margin (Shaw, 1977). However, the dark, largely bubble-free ice and absence of coarse-grained debris in the Western Margin basal ice facies suggests that the apron entrainment model may not describe its mode of formation. Case studies have demonstrated that basal ice in cold-based systems can also be produced by subglacial processes including the deformation and entrainment of subglacial permafrost (Fitzsimons et al., 2008), the overriding of ice marginal lakes (Lorrain et al., 1999), and the refreezing of water produced in warm thermal zones and/or high pressure zones at the glacier bed that then flows into cold thermal zones and/or low pressure zones downstream, where it refreezes (Knight, 1997; Wettlaufer et al., 1996) and entrains debris and excludes gases as it accretes to the glacier sole (Gilpin, 1979; Walder, 1986).

## 4.2 Chemistry

'Warm' basal ice from fast-flowing glaciers was consistently enriched in rock-derived solutes, including $SiO_2$, $SO_4^{2-}$, $K^+$, $Ca^{2+}$, $Mg^{2+}$, and $HCO_3^-$, compared to overlying glacier ice (Table 1, Fig. 3). These solutes were likely derived from weathering of reactive minerals such as carbonates, sulphides and aluminosilicates in the underlying bedrock (Tranter et al., 1996) that are commonly present in trace amounts (Holland, 1978). Furthermore, the presence of water during the formation of 'warm' basal ice would support reactions involving acid hydrolysis, which are usually the most important subglacial weathering processes (Raiswell, 1984). In contrast, limited rock-water contact during the formation and persistence of 'cold' basal ice likely limited chemical weathering, resulting in low mean solute concentrations (22 μeq/L) similar to those in glacier ice (15.6 μeq/L). Also like glacier ice, the solutes in 'cold' basal ice were dominated by atmospherically-derived components ($Cl^-$, $SO_4^{2-}$, and $Na^+$), rather than by solutes likely to be derived from the local sandstone, dolomite, limestone and conglomerate rocks ($Ca^{2+}$, $Mg^{2+}$, $HCO_3^-$).

## 4.3 Inorganic nutrients

Both 'warm' and 'cold' basal ice showed some evidence of inorganic nutrient acquisition in the subglacial environment. The 'cold' basal ice from the Western Margin had relatively high concentrations of TDP, TDN (particularly $NH_4^+$) (Table 1, Fig. 3) while the 'warm' basal ice samples were only occasionally enriched in reduced nitrogen ($NH_4^+$) and dissolved phosphorus (TDP) (Table 2). The substrate surrounding the Western Margin sample sites is composed largely of Cambrian and Ordovician sandstone, dolomite, limestone, and conglomerate (Harrison et al., 2016), which likely contain higher phosphorus



concentrations than do the metasedimentary rocks (Porder and Ramachandran, 2013) that underly the polythermal glaciers in this study. Phosphorus in rocks is almost exclusively found in apatite mineral groups (Taylor and McClennan, 1985) while nitrogen can occur in recalcitrant organic matter or $NH_4^+$ in silicate minerals (Honma, 1996; Honma and Schwarcz, 1979). The

distribution of these components, and thus P and N, in shield rocks, volcaniclastics and the Canadian Shield can be spatially heterogeneous  (Honma, 1996; Honma and Schwarcz, 1979). 'Warm' basal ice may also have variable inorganic nutrient concentrations if the location of subglacial biogeochemical activity is temporally and/or spatially heterogeneous. Subglacial microbial communities may function as a source of $NH_4^+$ via catabolic processes and the degradation of organic matter. Excess $NH_4^+$ would be particularly prevalent during the degradation of nitrogen-rich organic matter, such as the protein-like DOM

that was observed in basal ice (described by PARAFAC C1 and C2). In subglacial environments where organic matter contains insufficient nitrogen to meet the requirements for anabolic metabolism, specific organisms may fix nitrogen *in situ*. Genes indicative of nitrogen fixation (nitrogenase iron protein (*nifH*) gene) have been found in subglacial microbial communities (Boyd et al., 2011). However, despite the presence of *nifH* genes (Boyd et al., 2011) and observations of $N_2$ fixation on glacier surfaces (Telling et al., 2011), $N_2$ fixation has yet to be documented in subglacial environments.

**4.4 DOM**

Both 'warm' and 'cold' basal ice contained higher average DOC concentrations (0.49 ppm and 0.40 ppm, respectively) than glacier ice (0.15 ppm) (Table 1). Compared to glacier ice, the DOM in 'warm' and 'cold' basal ice had higher and more variable proportions of humic-like fluorescent material (C3 and C5) but no significant differences in the presence of C1, C2 or C4 protein-like fluorescent material (Table 1, Table 2, Fig. 3). Humic DOM, and humic-like C3 and C5 fluorescence are

commonly associated with soils and vegetation (Cory and McKnight, 2005; Osburn et al., 2016; Stedmon et al., 2003) so it is possible that both the fast and slow-flowing glaciers acquired these compounds from the overridden substrate during past glacial advances. In polythermal glaciers, high rates of mechanical weathering and meltwater contact with the underlying substrate could have facilitated the acquisition of humic-like DOM from the substrate.  Because the sedimentary rocks near/underlying the Western Margin support more well-developed soils and vegetation than the metasedimentary rocks

surrounding the fast-flowing polythermal glaciers in this study, even limited interaction with the substrate could have resulted in the acquisition of significant humic-like DOM in this cold-based system if this material was abundant in the substrate. Previous studies have also associated humic-like C3 and C5 fluorescence with microbial processing of organic matter (Table 2), suggesting basal ice may also have acquired these components via heterotrophic microbial activity in subglacial environments or in supraglacial or ice marginal material that was transported into the subglacial system by meltwater. The

positive correlation between DOC concentrations in 'warm' basal ice and tyrosine-like C1 fluorescence ($r_s$ = 0.61, p=0.02, n=9) indicates that a relatively consistent fraction of the DOC derived from these subglacial systems was proteinaceous in character and also suggests the presence of *in situ* subglacial microbial activity. The production of tyrosine-like fluorescence has been widely linked to the degradation of terrestrially-derived humic-like DOM and microbial exudates (Table 2). Since tyrosine-like fluorophores are considered to be highly biodegradable (Yamashita and Tanoue, 2003), it is likely that tyrosine-





like fluorescence was produced *in situ* within the subglacial environment from the degradation of allochthonous organic matter and/or the production of autochthonous organic matter. This is consistent with other studies that suggest subglacial environments contain both allochthonous organic matter (i.e. from bedrock/paleosols/overridden soils and vegetation) and autochthonous organic matter that may be produced *in situ* from microbial metabolism (Hodson et al., 2005; Wadham et al., 2016).


### 4.5 Microbial assemblages

The microbial assemblages contained in the 'cold' basal ice was remarkably similar to those in meteoric glacier ice; 'cold' basal ice shared most (i.e. 73%) OTUs with glacier ice, of which many (37%) were unique to only 'cold' basal ice and meteoric glacier ice (Fig. 4). Like glacier ice, the microbial assemblages observed in 'cold' basal ice included Proteobacteria

($\bar{x}$ = 42%), Bacteroidetes ($\bar{x}$ =16%), Actinobacteria ($\bar{x}$ =15%) and Cyanobacteria ($\bar{x}$ = 7.8%) (Table 1). Since microbial assemblages in soils and sediment typically differ considerably from those in the atmosphere and meteroric glacier ice, it is very unlikely that the substrate surrounding the Western Margin, composed of sandstone, dolomite, limestone and conglomerate rocks and relatively well-developed soils, is characterized by the same microbial assemblages as were observed in glacier ice. Therefore, the remarkable similarity between the microbial assemblages in 'cold' basal ice and glacier ice suggest

that either the 'cold' basal ice acquired few microbes from the underlying substrate during its formation or that most microbes acquired from the underlying substrate during its formation did not remain active *in situ* (and their DNA not preserved).

In contrast, the microbial assemblages in 'warm' basal ice from fast-flowing glaciers were distinct from those in glacier ice (Fig. 2, Fig. 4, MRPP, p<0.05). The 'warm' basal ice therefore probably acquired distinct microbes from the underlying substrate during its formation and may have also sustained *in situ* microbial activity, as has been reported in

laboratory incubation experiments of basal material (e.g. Boyd et al., 2011; Wadham et al., 2012). Like glacier ice, the microbial assemblages observed in 'warm' basal ice were dominated by proteobacteria ($\bar{x}$ =30%) and Actinobacteria ($\bar{x}$=30%), but they contained a significantly larger proportion of Chloroflexi ($\bar{x}$ =8.1%) and Gemmatimonadetes ($\bar{x}$ =3.9%) and a significantly smaller proportion of Bacteroidetes ($\bar{x}$ =9.2%) and Cyanobacteria ($\bar{x}$ =0.07%) than did glacier ice(Table 1). The 'warm' basal ice yielded a very small portion of 'ubiquitous' OTUs with less than 1% of the OTUs found in all 'warm' basal

ice samples. The abundances of these 'ubiquitous' subglacial organisms were also highly variable between samples, ranging between a minimum of <0.01% of the assemblage, to a maximum of 2-40% of the assemblages, suggesting that the distributions of either the source(s) of these microbes and/or their *in situ* activity (i.e. reproduction) are spatially heterogeneous. Thus, even though there is evidence for globally-distributed microbial species that are capable of survival across the range of extreme subglacial environments (Bhatia et al., 2006; Lanoil et al., 2009; Skidmore et al., 2005), these species appear to be

few and their local abundance may be highly dependent on site-specific conditions.

Although sample location did not affect the major ion chemistry, nutrient, or organic composition of basal ice, it was an important influence on the composition of the microbial assemblages in 'warm' basal ice. We observed that microbial assemblages in basal ice samples from the same glacier were more similar to each other than were assemblages in basal ice



from different glaciers. Inter-glacier differences in basal microbial assemblages were resolved over relatively small distances
(less than 100 km), and between glaciers with similar basal thermal regimes and underlying substrates (Fig. 1). Geographic
location has previously been identified as an important determinant of microbial assemblages across various spatial scales,
from meters (Lear et al., 2014) to global (Fuhrman et al., 2008), and within other polar environments including Antarctic and
Arctic terrestrial and aquatic habitats (Comte et al., 2016; Yergeau et al., 2007). Basal ice in different glaciers can be
particularly isolated from each other, so microbial dispersal between systems is probably very limited. Furthermore, although
residence times of 'warm' basal ice within a system are difficult to estimate, they may be sufficiently long to allow stochastic
processes, such as random extinction, chance colonization, drift, and priority effects (Chase and Myers, 2011; Vellend and
Agrawal, 2010) to play important roles in shaping the structure of microbial assemblages in basal ice. In contrast, basal
processes within a system, including ice deformation and melt-freeze effects, would provide some degree of intra-glacial
mixing of microbial assemblages and may explain the higher degree of similarity between assemblages in basal ice from the
same system.

## 5 Conclusions

We investigated the biogeochemical properties of 'warm' basal ice from three polythermal glaciers that drain a region of the
Devon Ice Cap that is underlain by metasedimentary rocks and gneiss. We found samples of basal ice from their subglacial
environments to be consistently enriched in solutes (i.e. $SiO_2$, $Na^+$, $K^+$, $Ca^{2+}$, $Mg^{2+}$, and $HCO_3^-$), a specific fraction of humic-
like fluorescent DOM (C3), and microbes compared to glacier ice of meteoric origin. Although these basal ice samples were
not enriched in nitrate, they were occasionally enriched in dissolved phosphorus (TDP), reduced nitrogen ($NH_4^+$) and a second
component of humic-like fluorescent DOM (C5), compared to glacier ice. The sources and/or sinks of these nutrients can
therefore be spatially heterogeneous in the relatively warm subglacial systems of polythermal glaciers. Large fractions of the
solutes, microbes, and nutrients derived from these subglacial systems were probably acquired directly from the underlying
substrate.

Unlike the biogeochemistry of the basal ice produced in relatively warm subglacial systems, basal ice produced in cold-based
systems retained characteristics remarkably similar to those of meteoric glacier ice. Although the 'cold' basal ice samples
analysed in this study may have acquired some inorganic and organic nutrients from the subglacial substrate, the substrate did
not appear to contribute significant solutes or microbes to the basal ice. It remains unknown whether the biogeochemical
differences between 'cold' basal ice and 'warm' basal ice reflect (i) differences in the characteristics of the
underlying/surrounding substrate, (ii) specific glaciological/hydrological processes that occurred during the formation of the
'cold' basal ice, or (iii) the effects of biogeochemical processes that occur *in situ* in 'cold' basal ice. Further research is required
to define how the 'cold' basal ice at the Western Margin of the DIC developed, and to better characterize the biogeochemical
processes that occur in subglacial environments where liquid water is limited. Nevertheless, findings from this study suggest



that basal temperature play important roles in controlling subglacial biogeochemistry and the suite of solutes, nutrients, and microbes that are either mobilized from the substrate or produced within subglacial systems.

## 5 Data availability

Sequences were submitted to the National Center for Biotechnology Information Sequence Read Archive. Other data from this paper are available upon request to the corresponding author.

## 6 Author Contributions


AD conceptualized the study, AD and BD designed and completed the fieldwork, AM completed the DNA extractions in the laboratory, AD completed formal analysis and wrote the manuscript with reviews and edits from all authors.

## 7 Competing interests

The authors declare that they have no conflict of interest.

## 9 Acknowledgements


This research was supported by an NSERC Discovery Grant and Northern Research Supplement to M. Sharp (05234-2015), Polar Continental Shelf Program (PCSP) for field support (604-16 and 620-17) and an NSTP grant, C/BAR grant, The W. Garfield Weston Award for Northern Research, and a NSERC CGSD Award to A. Dubnick. We thank the staff at the BASL Laboratory for chemical analyses, Chad Cuss at the SWAMP Laboratory for assistance with DOM/PARAFAC analyses, the
Neufeld lab at the University of Waterloo for DNA sequencing, and past field crews on the Devon Ice Cap for accumulating such a valuable wealth of knowledge about the ice cap.

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






**Table 1: Number, mean and standard deviation of measures of major ions, inorganic nutrients and DOM components in glacier ice, 'warm' basal ice, and 'cold' basal ice and sstatistical tests between 'warm' basal ice/'cold' basal and glacier ice. P-values that represent significant differences (p<0.05) are red.**

| | Units | Detection limit | Number | | | Mean | | | Standard Deviation | | | p-value | | | |
| | | | Glacier ice | 'Warm' Basal Ice | 'Cold' Basal Ice | Glacier ice | 'Warm' Basal Ice | 'Cold' Basal Ice | Glacier ice | 'Warm' Basal Ice | 'Cold' Basal Ice | 'Warm' BI vs glacier ice | | 'Cold' BI vs glacier ice | |
| | | | | | | | | | | | | T-test | F-test | T-test | F-test |
|---|---|---|---|---|---|---|---|---|---|---|---|---|---|---|---|
| **Ionic strength** | µeq/L | N/A | 11 | 12 | 5 | 15.6 | 241 | 22.0 | 7.13 | 265 | 10.4 | 0.00 | 0.00 | 0.17 | 0.30 |
| **SiO$_2$** | ppm | 0.02 | 11 | 12 | 5 | 0.04 | 0.24 | 0.04 | 0.01 | 0.31 | 0.00 | 0.01 | 0.00 | 0.75 | 0.00 |
| **Cl$^-$** | µeq/L | 0.85 | 11 | 12 | 5 | 2.92 | 9.10 | 5.25 | 1.14 | 16.5 | 2.30 | 0.17 | 0.00 | 0.02 | 0.07 |
| **SO$_4^{2-}$** | µeq/L | 0.83 | 11 | 12 | 5 | 3.60 | 19.6 | 4.33 | 3.09 | 25.8 | 3.69 | 0.33 | 0.17 | 0.68 | 0.59 |
| **Na$^+$** | µeq/L | 0.87 | 11 | 12 | 5 | 2.97 | 45.9 | 3.57 | 1.94 | 101 | 1.72 | 0.01 | 0.37 | 0.56 | 0.88 |
| **K$^+$** | µeq/L | 0.26 | 11 | 12 | 5 | 0.34 | 9.04 | 0.50 | 0.24 | 7.60 | 0.45 | 0.00 | 0.00 | 0.81 | 0.39 |
| **Ca$^{2+}$** | µeq/L | 0.50 | 11 | 12 | 5 | 2.31 | 43.3 | 2.49 | 1.36 | 53.0 | 2.39 | 0.00 | 0.00 | 0.85 | 0.14 |
| **Mg$^{2+}$** | µeq/L | 0.82 | 11 | 12 | 5 | 1.43 | 22.3 | 2.97 | 0.83 | 18.3 | 2.57 | 0.00 | 0.00 | 0.09 | 0.00 |
| **HCO$_3^-$** | µeq/L | 0.87 | 11 | 12 | 5 | 0.52 | 91.8 | -0.05 | 4.61 | 104 | 7.68 | 0.00 | 0.00 | 0.85 | 0.17 |
| **TDP** | P µg/L | 0.2 | 11 | 12 | 5 | 1.82 | 13.7 | 3.80 | 0.06 | 35.5 | 3.03 | 0.08 | 0.00 | 0.03 | 0.00 |
| **SRP** | P µg/L | 0.9 | 11 | 12 | 5 | 1.00 | 11.9 | 3.20 | 0.33 | 32.6 | 2.77 | 0.27 | 0.26 | 0.00 | 0.03 |
| **TDN** | N µg/L | 7 | 11 | 12 | 5 | 44.8 | 44.3 | 134 | 16.0 | 24.9 | 138 | 0.96 | 0.17 | 0.03 | 0.11 |
| **NO$_2^-$+ NO$_3^-$** | N µg/L | 2 | 11 | 12 | 5 | 11.9 | 9.08 | 6.00 | 5.99 | 10.7 | 3.67 | 0.00 | 0.00 | 0.06 | 0.36 |
| **NH$_4^+$** | N µg/L | 3 | 11 | 12 | 5 | 24.6 | 23.6 | 90.2 | 8.81 | 17.4 | 110 | 0.86 | 0.04 | 0.03 | 0.04 |
| **DOC** | ppm | 1.8 | 11 | 12 | 5 | 0.15 | 0.49 | 0.40 | 0.06 | 0.59 | 0.25 | 0.12 | 0.61 | 0.00 | 0.68 |
| **DOM C1** | FI | N/A | 10 | 9 | 5 | 3.24 | 3.72 | 3.22 | 2.94 | 3.71 | 2.24 | 0.76 | 0.50 | 0.99 | 0.63 |
| **DOM C2** | FI | N/A | 10 | 9 | 5 | 5.27 | 6.40 | 3.28 | 4.27 | 4.41 | 1.25 | 0.58 | 0.91 | 0.33 | 0.03 |
| **DOM C3** | FI | N/A | 10 | 9 | 5 | 1.63 | 6.44 | 21.2 | 1.46 | 6.48 | 28.5 | 0.04 | 0.00 | 0.00 | 0.00 |
| **DOM C4** | FI | N/A | 10 | 9 | 5 | 2.96 | 4.69 | 2.74 | 2.39 | 3.49 | 0.94 | 0.22 | 0.28 | 0.85 | 0.09 |
| **DOM C5** | FI | N/A | 10 | 9 | 5 | 1.95 | 4.78 | 6.77 | 2.25 | 5.48 | 5.29 | 0.15 | 0.02 | 0.03 | 0.03 |
| **Acidobacteria** | % | N/A | 5 | 11 | 3 | 1.1 | 3.2 | 1.5 | 0.84 | 3.6 | 1.2 | 0.73 | 0.70 | 0.54 | 0.53 |
| **Actinobacteria** | % | N/A | 5 | 11 | 3 | 17 | 30 | 15 | 6.4 | 22 | 21 | 0.81 | 0.75 | 0.86 | 0.05 |
| **Bacteroidetes** | % | N/A | 5 | 11 | 3 | 14 | 9.2 | 16 | 5.2 | 15 | 18 | 0.00 | 0.03 | 0.82 | 0.04 |
| **Chloroflexi** | % | N/A | 5 | 11 | 3 | 0.7 | 8.1 | 6.1 | 0.51 | 5.6 | 9.1 | 0.01 | 0.00 | 0.20 | 0.00 |
| **Cyanobacteria** | % | N/A | 5 | 11 | 3 | 16 | 0.07 | 7.8 | 17 | 0.11 | 7.5 | 0.04 | 0.00 | 0.47 | 0.34 |
| **Firmicutes** | % | N/A | 5 | 11 | 3 | 1.0 | 10 | 0.06 | 1.9 | 15 | 0.10 | 0.10 | 0.24 | 0.54 | 0.22 |
| **Gemmatimonadetes** | % | N/A | 5 | 11 | 3 | 0.39 | 3.9 | 0.02 | 0.23 | 4.9 | 0.02 | 0.01 | 0.01 | 0.04 | 0.01 |
| **Proteobacteria** | % | N/A | 5 | 11 | 3 | 43 | 30 | 42 | 16 | 17 | 9.1 | 0.17 | 0.99 | 0.92 | 0.52 |







**Table 2: Excitation and emission maxima for the 5 component PARAFAC model, including the identification of each component**

| | Ex (nm) | Em (nm) | Description | # of OpenFlour matches* |
|---|---|---|---|---|
| C1 | 276 | 300 | **Protein (tyrosine)-like fluorescence** that may originate from the degradation of terrestrially-derived humic-like DOM (Coble, 2007; Coble et al., 1998; Mopper and Schultz, 1993; Stedmon and Markager, 2005) and microbial exudates (Smith et al., 2017). | 0 |
| C2 | 230 (286) | 325 | **Protein (tryptophan)-like fluorescence** (Coble et al., 1998; Lakowicz, 1999) that has been liked to microbial activity (Elliott et al., 2006) and is associated with the autochthonous production of DOM in various environments (Coble, 1996; Fellman et al., 2008; Mopper and Schultz, 1993; Yamashita and Tanoue, 2003) | 3 |
| C3 | 232 (336) | 460 | **Humic (fulvic acid)- like fluorescence** derived from higher plants (terrestrial) and/or organic matter with a certain degree of microbial processing (Cory and McKnight, 2005; Osburn et al., 2016) | 10 |
| C4 | 298 (231) | 410 | **Humic-like fluorescence** that can be highly processed terrestrial DOM with low biolability (Lapierre and Del Giorgio, 2014) | 0 (4**) |
| C5 | 237 (317) | 395 | **Humic-like fluorescence** from marine (Coble, 1996) and terrestrial (Stedmon et al., 2003) environments and can be affiliated with microbial reprocessing of organic matter (Stedmon and Markager, 2005) | 12 |

\* >95% certainty
\*\* 90-95% certainty

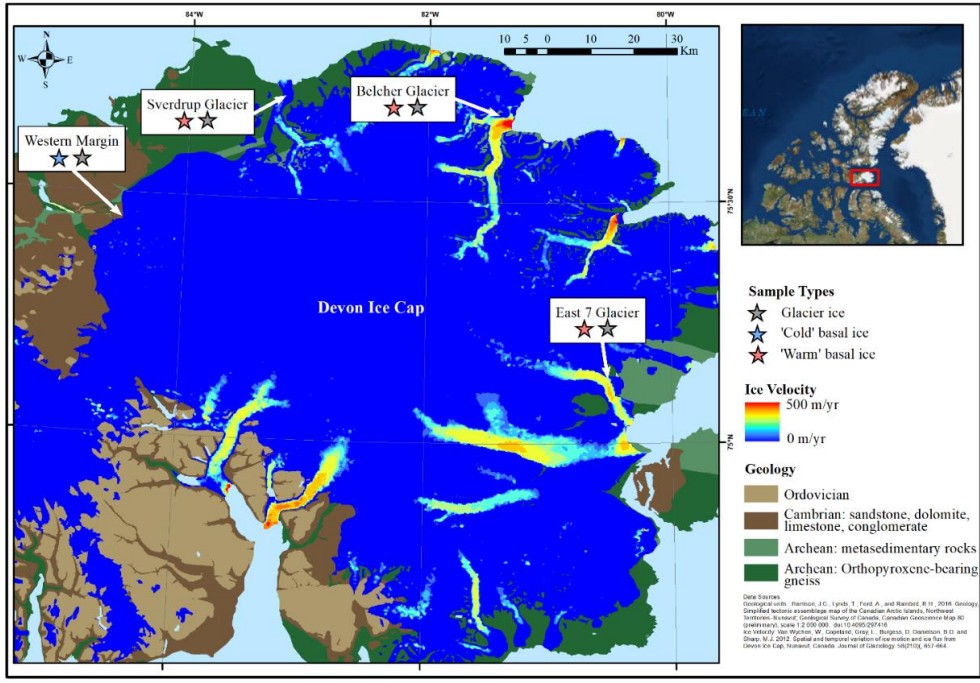


**Figure 1: Study Site indicating the geology of the surrounding substrate [*Harrison et al.*, 2016] and flow velocity of the Devon Ice Cap [*Van Wychen et al.*, 2012]. Samples were collected from 3 polythermal glaciers with relatively fast flowing ice that are surrounded by Archean bedrock, and two locations along the relatively slow flowing cold-based section of the Western Margin.**



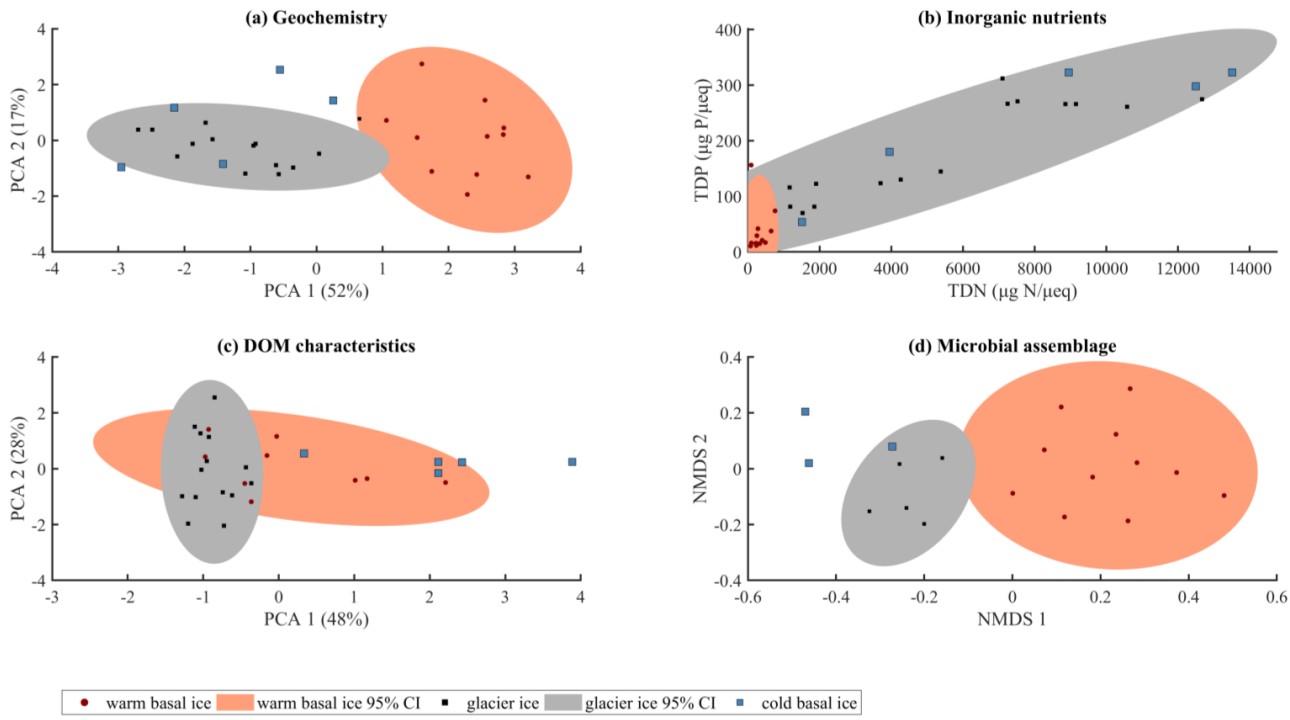

**Figure 2: Summary of a) major ion composition, produced by PCA using the contribution of each major ion to the solute load, with each major ion normalized to its mean and variance b) inorganic nutrients relative to the solute load (TDP (µg P/µeq) and TDN (µgN/µeq)) (c) character of dissolved organic matter determined by principle component analysis using the relative contributions of the 5 modelled fluorescent components, with each component normalized to its mean and variance and d) microbial assemblage structure determined by nonmetric multidimensional scaling (NMDS) of Bray-Curtis distance measure using 16S rRNA gene sequencing (stress=0.12).**


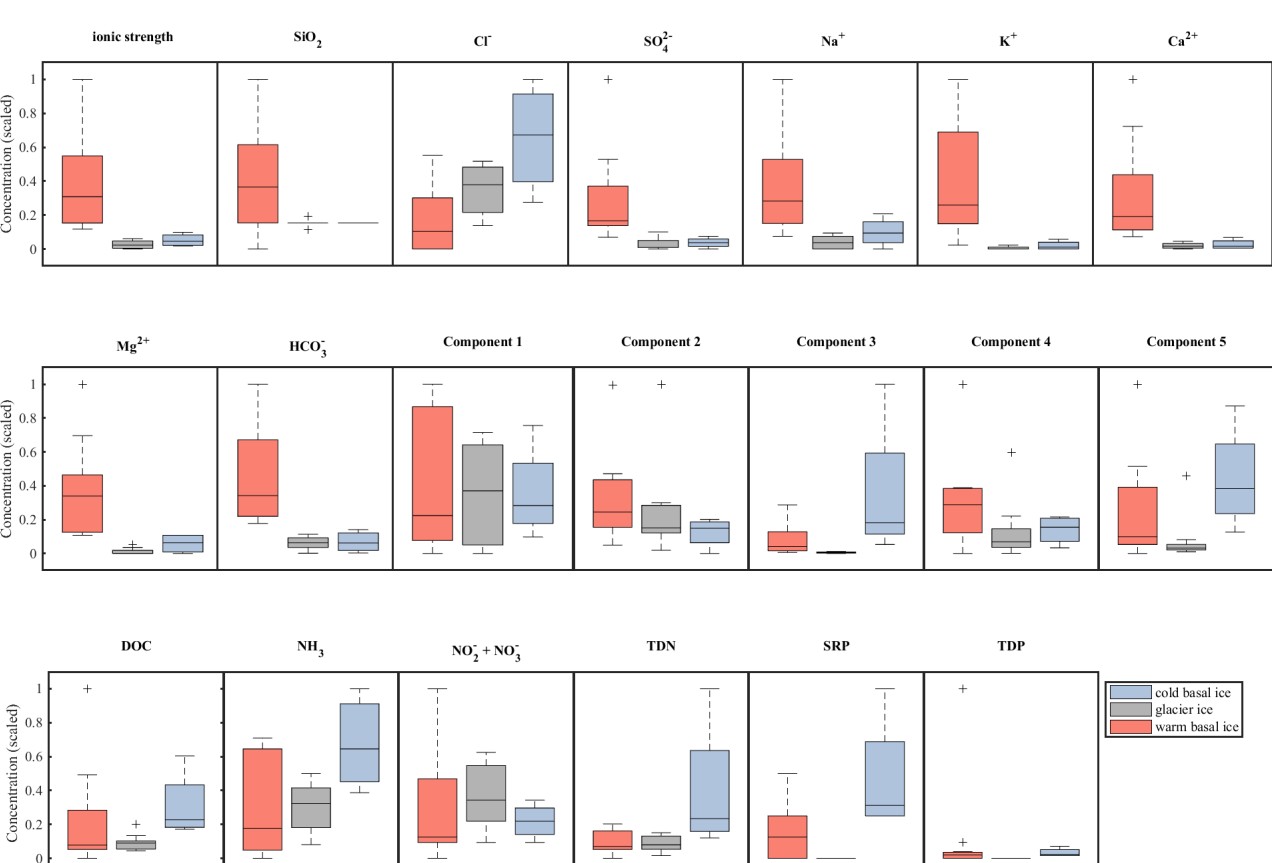

**Figure 3: Relative abundance and range in concentrations of major ions (top), organic nutrients (middle) and inorganic nutrients (bottom) in basal ice and glacier ice. Data were scaled to the interval 0-1 and boxplots indicate the median, 25th and 75th percentiles, whiskers indicate the most extreme datapoints not considered outliers and outliers are indicated with a '+' symbol.**






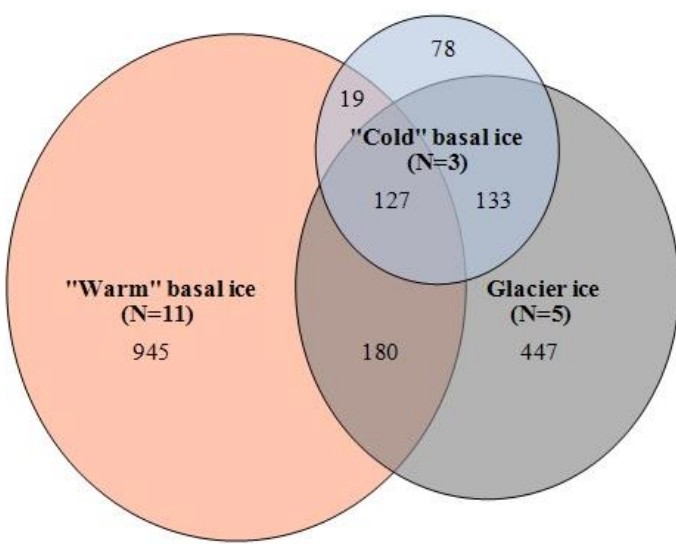

**Figure 4: Venn Diagrams showing overlap in membership between the microbial assemblages observed in 'warm' basal ice, glacier ice, and 'cold' basal ice samples. Numbers represent the number of operational taxonomic units (OTUs) that are unique to each environment or shared between environments.**