# Peer review of "Basal thermal regime affects the biogeochemistry of subglacial systems"

_Biogeosciences, 2019_

## Referee Comment (RC1) · Marek Stibal (Referee) · 18 Sep 2019

The paper "Basal thermal regime affects the biogeochemistry of subglacial systems" by Ashley Dubnick and co-authors tests the hypothesis that glacier thermal regime controls subglacial biogeochemical processes through bed(rock) material mobilisation. The authors collected samples of basal and meteoric ice from one cold-based and three polythermal glaciers of the Devon Ice Cap in the Canadian Arctic and compared their solute and nutrient contents and microbial communities. They found that while basal ice was enriched in solutes, nutrients, and microbes compared to the respective overlying ice in polythermal glaciers, this was not the case in the cold-based glacier investigated. Moreover, location seemed to play a more important role for microbial community structure than ice type.

To my knowledge this is the first study explicitly addressing the role of thermal regime on subglacial biogeochemistry and as such it is a very valuable contribution to glacial ecosystem research. The sites were carefully selected and sampled, adequate methods were used to analyse their solute and nutrient contents and microbial community structure, and the conclusions are well-supported by the results. The ms is logically structured and well written.

Overall, this is an interesting study suitable for BG. I have a few comments/criticisms I would like the authors to address before the ms is published though.

First, the results of microbiological analysis should be showed and discussed in more detail (e.g. how many raw sequences were obtained and how did that change after rarefaction; how many OTUs were identified and were the dominant OTUs similar to those of other glacial environments; was microbial abundance in the samples quantified in any way?).

Second, it is a bit unfortunate that the only cold-based glacier sampled had a different bedrock type than the three polythermal glaciers, as it makes the differences between the sites more difficult to explain (bedrock vs. thermal regime effect). This should be acknowledged in the relevant sections of the discussion.

Last, there is a discrepancy between DOC (both warm and cold basal ice contained more DOC, including proteinaceous material, compared with meteoric ice) and microbial communities (warm basal ice vs. cold basal ice and meteoric ice). This is in my opinion not sufficiently explained in the ms. Is it because solutes are entrained even by cold-based glaciers but particulates are not? Or may it be an effect of bedrock (see above)?

Minor comments

260-264 As you didn't specifically look at any microbial functions, the discussion of potential N2 fixation feels a bit out of place here and could be deleted.
266-289 Emily O'Donnell (Lawson)'s 2016 in BG was the first detailed study on DOM in basal ice and showed e.g. the importance of bedrock and leaching of DOM in wet conditions at the glacier bed. I think it would be a useful reference for this section.

305 Here, the 2012 Global Change Biology paper would be a more appropriate reference, as the experimental data in Wadham et al. come from it (as the first author of this ms surely remembers...).

323 There already exist spatially explicit studies of microbial communities in glacial environments, mostly the surface – e.g. Cameron et al. 2016 FEMS, Darcy et al. 2017 FEMS. We also found spatial differences in Disko Island glacier stream assemblages (Zarsky et al. 2018 FEMS). These studies might be worth mentioning here.

Figure 3 seems to show data already shown in Table 1. If this is the case it may be redundant.

[Figure]

---

## Referee Comment (RC2) · Anonymous Referee #2 · 25 Sep 2019

**Anonymous Referee #2**

This article by Ashley Dunnick and co-authors considers basal thermal regime as a governing mechanism for acquisition of solutes and microbial biomass into the basal ice. Somewhat unsurprisingly, basal thermal regime plays an important role in subglacial biogeochemical processes. Through a comparison of the basal ice from three polythermal and one cold based glacier, to the respective overlying ice, it is found that cold based glacier's basal ice is of similar composition to its overlying ice, while polythermal basal ice is enriched with solutes derived from the substrate material, microbes and metabolic products. It is highly warranted and useful that the comparison between cold and polythermal basal ice has been quantified. To my knowledge, this is the first study that explicitly makes this comparison.

The strengths of this paper lie in the meticulous sampling strategy, where overlying

and basal ice were carefully delineated and distinguished. Furthermore, this study uses appropriate field sampling and analytical techniques. It would have been useful to compare between a cold based glacier with similar bedrock geology to the three polythermal glaciers. Nonetheless, the study still makes an interesting contribution.

Overall, the paper is well-structured. However, I have a couple of major concerns about the sample analysis and findings. Furthermore, I have a few comments on the writing style and formatting.

Methods:

1) The analytical methods, as presented, are relatively sparse, especially in relation to the analytical procedure for SRP, TDN and TDP. It would be useful to include a description of the digests performed and the recovery. It would also be useful to understand if any reference material or standards were used and the outcome of this.

Findings:

2) The concentrations reported for DOC in Table 1 appear to be less than the LoD, in numerous cases. Additionally, there is a discrepancy between the LoD cited in text and in the table. Naturally, it is highly problematic if the concentrations reported are lower than the LoD. The statistical differences and comparison between basal and overlying ice, referred to in the results and discussion, would have to be amended. If authors wish to keep DOC data included in this exercise, they need to make it clear to the reader that their DOC data is of good quality, by providing transparent details on the methodology, as well as appropriate use of CRMs.

Writing Style/ Formatting:

3) The abstract and introduction's wording should be tightened to maintain clarity and flow. There is a considerable number of lists. Often lists of factors, studies or processes are lengthy and the point can become lost. Amending this will help the research aim (in line 46) to be stated more clearly. Currently, the importance of this line is lost. The

meaning is lost elsewhere, for example in lines 28 through to 31.

4) Additional reference to important literature, especially that relating to microbially mediated chemical weathering, could be made in the introduction. Similarly, the discussion is sparsely referenced, particularly in the first paragraph.

5) In the introduction, it would be useful to include a few additional lines on the importance of this study. Why should the broad readership of BG care about this study? I know that the majority of this study may be lost on the readership, due to the intricacy of the comparison specifically relating to glacial systems. As such, the importance of this work for the BG's audience needs to be clarified.

6) There is some repetition in the methodology and introduction– especially in relation to the field sampling and the definition of warm/cold basal ice.

7) Table 1 is rather lengthy and unclear, it may be useful to split the table up into its component parts (chemistry, in/organic nutrients and microbes) or to reformat the table.

Technical corrections/comments:

1) Is the use of 'warm' and 'cold' in quotations necessary throughout? I feel it is not, as long as you state early on that these are the terms you are going to use.

2) The definition of cold based and warm based glaciers is repeated throughout the paper – it is only really necessary to define these terms once.

3) Consider revising the word 'parent' in 'parent ice' - this could lead to inference that the basal ice is always of younger age, which is not necessarily true. As such, this phrase may be slightly misleading. Consider revising throughout. If you choose to use parent ice – this should be defined and used consistently.

4) There are many sentences which are poorly constructed, with use of multiple 'and/s', 'and/or' and 'also/s'. Often, this disrupts clarity and flow. Please consider revising.

5) Line 32 - are you missing a reference related to subglacial microbial mediated chemical weathering?

6) Consider rephrasing line 39 for clarity – the first sentence is a little unclear, I think you may be missing a word.

7) Line 44  274 - too many spaces.

8) Line 259, this would be an appropriate place to reference the Wadham (2016) study.

9) Line 274 - consider rephrasing the sentence starting with 'Because. . .'.

10) Line 289, although production and consumption of autochthonous OM are mentioned in Wadham (2016), I think there are other more appropriate references for this point.

11 ) The font size of the figure captions vary.

12) Please, standardized units throughout. For example, currently, there is a mixing of DOC units mg L-1 and ppm.
* * *

---

## Referee Comment (RC3) · Anonymous Referee #3 · 3 Oct 2019

The authors of this manuscript evaluate the effect of basal thermal regime on the characteristics of effluent from 3 glaciers associated with and one edge of the Devon Ice Cap. The goal was to see if the differences in movement and basal thermal regime resulted in differences in solutes, dissolved organic matter composition, and microbial community composition as assessed by comparing 16S amplicons from each site. They found the three glacial sites (warm basal ice) to be different from the one cold basal site taken at the western edge of the ice cap. The authors hypothesize that "basal thermal regime plays an important role in defining the physical and biogeochemical characteristics and variability of basal ice"; although one could argue that a hypothesis that states - differences in temperature affect microbial assemblages, weathering , and biogeochemical processes is more of a null hypothesis than an alternative hypothe-

sis. More importantly though, because they have three warm sites and a single cold site it seems that this hypothesis is untestable and that the statistical comparisons between each feature (warm basal ice vs. cold basal ice) relies on variation from multiple samples taken with each site and not from truly replicated glacier characteristics in the landscape (i.e. pseudoreplication). For the warm basal site, I think it is reasonable to say that there are triplicate samples (i.e., n=3), however for the cold basal ice site appears to be unreplicated (i.e., n=1) and so that hypothesis cannot be tested using the standard statistical analyses employed (t-tests, table 1). That said I wouldn't consider that a fatal flaw in the manuscript as what they are reporting is primarily observational and exploratory in nature and there is also value in that. Beyond this principal concern the paper is well-written, clear and straight forward. The comparisons among effluent chemistry, DOM fluorescent properties, and 16S amplicons are standard and do a reasonable job describing the differences among sampling locations. Given how hard these samples are to gather, how quickly the planet is losing the cryosphere, and how little we understand about the characteristics of glacier effluent being released different glacier types, it seems that these data, presented as they are in a clear and unambiguous fashion, are valuable and merit publication.

Very Minor Comments:

Line 64: I am not sure what the "a" refers too in "> 20 m a-1" but perhaps this is a common unit from studies on glaciers that I am unfamiliar with

Line 199 - this seems like an odd way to report this. "less than half a percent of the OTUs" perhaps <0.5% would be clearer?

---

## Author Comment (AC1) · 27 Nov 2019

**We would like to thank Marek Stibal for taking the time to review the manuscript and for providing very relevant and constructive comments – please find responses to each below.**

First, the results of microbiological analysis should be showed and discussed in more detail (e.g. how many raw sequences were obtained and how did that change after rarefaction; how many OTUs were identified and were the dominant OTUs similar to those of other glacial environments; was microbial abundance in the samples quantified in any way?).

Response: Microbial abundance in the samples were not quantified. We have included a sentence in Methods/Data Processing and Statistical Analyses to include the min (7,608) and max (188,117) number of raw reads per sample and the number of reads after rarefaction (7,608). We also added to the Results/Microbial assemblage the total number of OTUs identified across the rarefied dataset (3,555 OTUs) and material was added to the Discussion/Microbial assemblage to relate dominant OTUs to those identified in other glacial environments and prior literature: "Like glacier ice, the microbial assemblages observed in cold basal ice included Proteobacteria ($\bar{x}$ = 42%), Bacteroidetes ($\bar{x}$ =16%), Actinobacteria ($\bar{x}$ =15%) and Cyanobacteria ($\bar{x}$ = 7.8%) (Table 1). Proteobacteria, Bacteroidetes, Actinobacteria, and Cyanobacteria commonly dominate the microbiome of surface environments such as cryoconite holes (e.g. Cameron et al., 2012), glacier ice (e.g. Christner et al., 2005) and snow (e.g. Harding et al., 2011), and Cyanobacteria, Proteobacteria, and Actinobacteria contain organisms with the potential to photosynthesise (Cameron et al., 2012)." […] "The microbial assemblages observed in warm basal ice were dominated by Proteobacteria ($\bar{x}$ =30%) and Actinobacteria ($\bar{x}$=30%) which are commonly observed in cold ecosystems (Amato et al. 2007; Møller et al. 2013) including those in glacier ice (this study), basal ice (Stibal et al., 2012b; Yde et al., 2010), and subglacial waters (Christner et al., 2006; Rondón et al., 2016). Additionally, the warm basal ice contained a large proportion of Chloroflexi ($\bar{x}$ =8.1%) and Gemmatimonadetes ($\bar{x}$ =3.9%), which are common and active in permafrost soils (Tuorto et al., 2014) and other basal ice environments (Yde et al., 2010) but were significantly less dominant in glacier ice samples in this study (Table 1; T-test, $p<0.05$). Warm basal ice also contained relatively few Bacteroidetes ($\bar{x}$ =9.2%) and Cyanobacteria ($\bar{x}$ =0.07%) which were more abundant in glacier ice (Table 1) and in other surface environments (Cameron et al., 2012; Harding et al., 2011)."

Second, it is a bit unfortunate that the only cold-based glacier sampled had a different bedrock type than the three polythermal glaciers, as it makes the differences between the sites more difficult to explain (bedrock vs. thermal regime effect). This should be acknowledged in the relevant sections of the discussion.

Response: Agreed that this is unfortunate and a significant limitation to the study, which is why we attempted to frame the discussion of the warm and cold basal ice by comparing them to meteoric glacier ice rather than directly to each other. Conveniently, the cold basal ice remained remarkably similar to meteoric glacier ice even though it was surrounded by/overlying a more reactive substrate (sandstone, dolomite and limestone) than the warm-based systems (metasedimentary rocks and gneiss). Thus, the warm basal ice acquired more material from a

relatively unreactive substrate while the cold basal ice acquired little material from a reactive substrate. Material on this topic was already integrated in the discussions of "Inorganic nutrients", "DOM" and "Microbial Assemblages" but additional text was added to the Discussion under 'Chemistry' : "The cold basal ice therefore contained solute concentrations and compositions more similar to those in meteoric glacier ice than warm basal ice, even though the substrate beneath the cold-based ice (local sandstone, dolomite, limestone and conglomerate substrate) is likely far more reactive than the substrate beneath the warm-based glaciers (metasedimentary rocks and gneiss). Therefore, the differences in chemistry between these basal environments cannot be explained by differences in substrate alone."

The text regarding cold basal ice in the conclusion was rewritten to better highlight that the differences in biogeochem can not be fully explained by differences in substrate composition alone: "While basal ice in warm subglacial systems appear to have acquired abundant solutes, microbes and nutrients from the underlying substrate, basal ice produced in cold-based systems acquired few biogeochemical characteristics from the underlying substrate. The cold basal ice explored in this study may have acquired some inorganic and organic nutrients from the substrate, but acquisition of other solutes or microbes appear to be limited. This cold basal ice acquired few solutes and microbes even though the local substrate, composed of sandstone, dolomite and limestone, and relatively well developed soils, would have been more reactive than the metasedimentary and gneiss substrate beneath the warm-based systems."

Last, there is a discrepancy between DOC (both warm and cold basal ice contained more DOC, including proteinaceous material, compared with meteoric ice) and microbial communities (warm basal ice vs. cold basal ice and meteoric ice). This is in my opinion not sufficiently explained in the ms. Is it because solutes are entrained even
by cold-based glaciers but particulates are not? Or may it be an effect of bedrock (see above)?

Response: This research indicates that basal processes in warm-based glaciers result in the acquisition of abundant solutes, microbes and nutrients from the substrate (and possibly from in situ processes), but that cold basal ice appeared to have only acquired specific nutrients from the substrate and not microbes or bulk solutes (even though the substrate was more reactive). We can therefore argue that basal temperature likely plays an important role in controlling subglacial biogeochem. However, identifying the detailed processes that result in the biogeochemical intricacies (and contradictions) of cold-basal ice (ie DOC vs microbes) is beyond the scope of this study - we do not have sufficient information to identify detailed substrate characteristics, mechanism(s) of basal ice formation, or in situ processes that may occur so even speculating in any detail would be unconstrained, and as far as we are concerned, there is no obvious explanation. Instead, we have tweaked the last two sentences of the conclusion to explicitly highlight this the gap in our collective understanding of cold-basal ice biogeochem (and that at this field site) and suggest further research be conducted to help answer these pending and fundamental questions that this research has highlighted: "It remains unknown whether the intricacies of the biogeochemical characteristics that were observed in the cold basal ice in this study result from (i) specific characteristics of the underlying/surrounding substrate, (ii) specific

glaciological/hydrological processes that occurred during the formation of the cold basal ice, or (iii) the effects of biogeochemical processes that occur *in situ* in cold basal ice. Further research is required to define how the cold basal ice at the Western Margin of the DIC developed, and to better characterize the biogeochemical processes that occur in subglacial environments where liquid water is limited."

Minor comments
260-264 As you didn't specifically look at any microbial functions, the discussion of potential N2 fixation feels a bit out of place here and could be deleted.
Response: Agreed, so this discussion regarding potential N2 fixation was deleted

266-289 Emily O'Donnell (Lawson)'s 2016 in BG was the first detailed study on DOM in basal ice and showed e.g. the importance of bedrock and leaching of DOM in wet conditions at the glacier bed. I think it would be a useful reference for this section.
Response: Agreed. The paragraph regarding DOM substrate and microbial DOM sources was revised/expanded to integrate Lawson et al's (2016) findings "Both warm and cold basal ice contained higher average DOC concentrations (0.49 ppm and 0.40 ppm, respectively) than glacier ice (0.15 ppm) (Table 1) suggesting a potential source of DOC in subglacial systems, as observed in Greenland (Lawson et al., 2016) and Antarctica (Wadham et al., 2012)" and "Humic DOM, and humic-like C3 and C5 fluorescence are commonly associated with soils and vegetation (Cory and McKnight, 2005; Osburn et al., 2016; Stedmon et al., 2003) so it is possible that both the fast and slow-flowing glaciers acquired these compounds by direct (via abiotic leaching) and indirect (via microbial cycling) of material from the substrate. Similar observations were made  for low molecular weight DOC compounds in previous studies of basal ice from Greenland (Lawson et al., 2016)."

305 Here, the 2012 Global Change Biology paper would be a more appropriate reference, as the experimental data in Wadham et al. come from it (as the first author of this ms surely remembers: : :).
Response: Sincere apologies to the author of this ms – it has been included instead!

323 There already exist spatially explicit studies of microbial communities in glacial environments, mostly the surface – e.g. Cameron et al. 2016 FEMS, Darcy et al. 2017 FEMS. We also found spatial differences in Disko Island glacier stream assemblages (Zarsky et al. 2018 FEMS). These studies might be worth mentioning here.

Response: Thanks for the suggestion – the paragraph was revised to include reference to these studies: "Geographic location has previously been identified as an important determinant of microbial assemblages across various spatial scales, from meters (Lear et al., 2014) to global (Fuhrman et al., 2008), and within other polar environments including Antarctic and Arctic terrestrial and aquatic habitats (Comte et al., 2016; Yergeau et al., 2007) as well as on glacier surfaces (Cameron et al., 2016) and in subglacial discharge (Zarsky et al., 2018)."

Figure 3 seems to show data already shown in Table 1. If this is the case it may be redundant.
Response: Table 1 contains a summary of important and necessary statistical results that are heavily referenced throughout the text. However, it contains a lot of condensed information so is

perhaps difficult for some readers to navigate and discern trends (see comment #7 from Reviewer #2). Therefore Figure 3 was included to provide a visual summary (of scaled results) from which trends can be more easily discerned for the average reader.

---

## Author Comment (AC2) · 27 Nov 2019

**We would like to thank this anonymous reviewer for their time and detailed review of this manuscript – please find responses to each below.**

Methods:
1) The analytical methods, as presented, are relatively sparse, especially in relation to the analytical procedure for SRP, TDN and TDP. It would be useful to include a description of the digests performed and the recovery. It would also be useful to understand if any reference material or standards were used and the outcome of this.

Response: SRP was measured directly as $PO_4^{3-}$ while TDP was digested with potassium persulfate to convert all dissolved P to $PO_4^{3-}$. TDN was digested with potassium persulfate and sodium hydroxide to convert all dissolved N to $NO_3^-/NO_2^-$. Analyses were conducted in an ISO17025 accredited laboratory and reference material and standards were applied according to those standards. These details have been added to Methods/Analytical Methods.

Findings:
2) The concentrations reported for DOC in Table 1 appear to be less than the LoD, in numerous cases. Additionally, there is a discrepancy between the LoD cited in text and in the table. Naturally, it is highly problematic if the concentrations reported are lower than the LoD. The statistical differences and comparison between basal and overlying ice, referred to in the results and discussion, would have to be amended. If authors wish to keep DOC data included in this exercise, they need to make it clear to the reader that their DOC data is of good quality, by providing transparent details on the methodology, as well as appropriate use of CRMs.

Response: DOC detection limits were incorrectly reported and have been revised to 0.06 ppm throughout. Five standards between 0 ppm and 2 ppm were used for calibration ($R^2=1.0$) and the LoD was calculated based on instrument blanks according to methods outlined by Shrivastava and Gupta (2011). These details were added to the Analytical Methods/DOC concentrations.

Writing Style/ Formatting:
3) The abstract and introduction's wording should be tightened to maintain clarity and flow. There is a considerable number of lists. Often lists of factors, studies or processes are lengthy and the point can become lost. Amending this will help the research aim (in line 46) to be stated more clearly. Currently, the importance of this line is lost. The meaning is lost elsewhere, for example in lines 28 through to 31.

Response: The abstract and introduction have been revised and tightened and detailed lists were removed.

4) Additional reference to important literature, especially that relating to microbially mediated chemical weathering, could be made in the introduction. Similarly, the discussion is sparsely referenced, particularly in the first paragraph.

Response: We added Wadham et al (2004) as another example of microbial mediated redox reactions at the bed of glaciers in the Introduction. We also added Price and Sowers (2004) and Hubbard et al (2009) references to the first paragraph of the Discussion/Basal ice formation and a few other references throughout the Discussion including: O'Donnel et al (2016) in reference to subglacial DOM, Cameron et al (2012), Christner et al. (2005), Harding et al., (2001), Yde et al., (2010), Stibal et al (2012)., Rondon et al. (2016) and Tuorto et al. (2014) in reference to microbial

assemblages, and Cameron et al (2016) and Zarsky et al (2018) in reference to geographic influence on microbial assemblages. An effort was made to reference review papers and initial seminal research throughout the presentation of high-level interdisciplinary concepts to maintain readability.

5) In the introduction, it would be useful to include a few additional lines on the importance of this study. Why should the broad readership of BG care about this study? I know that the majority of this study may be lost on the readership, due to the intricacy of the comparison specifically relating to glacial systems. As such, the importance of this work for the BG's audience needs to be clarified. Response: The first and second paragraphs of the introduction were revised to more clearly articulate the importance of this study to the BG community:

"Glaciers form by the compression and metamorphism of snow and slowly deform and flow under their own weight. A considerable portion of a glacier's ice is of meteoric origin and receives chemical and biological inputs primarily from the atmosphere. However, subglacial processes, including melt-freeze events and erosion can result in the production of basal ice near the bed. This basal ice is typically characterized by relatively high concentrations of solutes that are dominated by $Ca^{2+}$, $Mg^{2+}$, $HCO_3^-$ and $SO_4^{2-}$ (Tranter, 2007). These solutes are often produced from reactions that involve carbonate and sulphide minerals (Tranter, 2007), which are trace components in most types of bedrock (Holland, 1978). Basal ice can also contain organic matter, nutrients (e.g. phosphorus, silica, potassium) and microbes from the underlying substrate (Montross et al., 2014; Sharp et al., 1999). Both basal ice and subglacial water are known to host populations of microbes that mediate redox reactions (e.g. Sharp et al., 1999; Wadham et al., 2004), play an active role in bedrock weathering (e.g. Tranter et al., 2002), and produce and/or consume ecologically important nutrients (e.g. Bottrell and Tranter, 2002; Boyd et al., 2011; Hodson, 2007; Statham et al., 2008; Tranter et al., 2002; Wadham et al., 2012)

Subglacial processes and the composition of basal ice can dramatically impact the biogeochemistry of meltwater and sediments exported from glaciers in a warming world. For example, in glaciers where surface-derived meltwater drains through the subglacial environment and comes into contact with basal ice, subglacial water and sediments, its geochemistry (Tranter et al., 2002), nutrient content (Hawkings et al., 2014; Wadham et al., 2016) and microbial community composition (Dubnick et al., 2017) are dramatically altered. Direct links have recently been established between subglacial biogeochemical signatures and impacts on downstream environments including downstream freshwater (Sheik et al., 2015) and fjord ecosystem (Gutiérrez et al., 2015). Similarly, during glacial retreat, the biogeochemical material contained in basal ice are released to the terrestrial landscape. These materials have been directly linked to the nutrient dynamics of glacier forefields (Kazemi et al., 2016; Mindl et al., 2007; Sattin et al., 2010) and form the basis of the soils from which many postglacial landscapes evolve (Kastovská et al., 2005)."

6) There is some repetition in the methodology and introduction– especially in relation to the field sampling and the definition of warm/cold basal ice. Response: Agreed, so repetition regarding field sampling and the definition of warm/cold basal ice was removed from the methods section.

7) Table 1 is rather lengthy and unclear, it may be useful to split the table up into its component parts (chemistry, in/organic nutrients and microbes) or to reformat the table.
Response: The table has been split into its component parts (chemistry, inorganic nutrients, organic nutrients and microbial assemblages):

**Table 1: Number, mean and standard deviation of measures of major ions, inorganic nutrients and DOM components in glacier ice, warm basal ice, and cold basal ice and sstatistical tests between warm basal ice/cold basal and glacier ice. P-values that represent significant differences (p<0.05) are red.**

| | Units | Detection limit | Number | | | Mean | | | Standard Deviation | | | p-value | | | |
| | | | Glacier ice | Warm Basal Ice | Cold Basal Ice | Glacier ice | Warm Basal Ice | Cold Basal Ice | Glacier ice | Warm Basal Ice | Cold Basal Ice | Warm BI vs glacier ice | | Cold BI vs glacier ice | |
| | | | | | | | | | | | | T-test | F-test | T-test | F-test |
|---|---|---|---|---|---|---|---|---|---|---|---|---|---|---|---|
| Chemistry | | | | | | | | | | | | | | | |
| Ionic strength | µeq/L | N/A | 11 | 12 | 5 | 15.6 | 241 | 22.0 | 7.13 | 265 | 10.4 | 0.00 | 0.00 | 0.17 | 0.30 |
| $SiO_2$ | ppm | 0.02 | 11 | 12 | 5 | 0.04 | 0.24 | 0.04 | 0.01 | 0.31 | 0.00 | 0.01 | 0.00 | 0.75 | 0.00 |
| $Cl^-$ | µeq/L | 0.85 | 11 | 12 | 5 | 2.92 | 9.10 | 5.25 | 1.14 | 16.5 | 2.30 | 0.17 | 0.00 | 0.02 | 0.07 |
| $SO_4^{2-}$ | µeq/L | 0.83 | 11 | 12 | 5 | 3.60 | 19.6 | 4.33 | 3.09 | 25.8 | 3.69 | 0.33 | 0.17 | 0.68 | 0.59 |
| $Na^+$ | µeq/L | 0.87 | 11 | 12 | 5 | 2.97 | 45.9 | 3.57 | 1.94 | 101 | 1.72 | 0.01 | 0.37 | 0.56 | 0.88 |
| $K^+$ | µeq/L | 0.26 | 11 | 12 | 5 | 0.34 | 9.04 | 0.50 | 0.24 | 7.60 | 0.45 | 0.00 | 0.00 | 0.81 | 0.39 |
| $Ca^{2+}$ | µeq/L | 0.50 | 11 | 12 | 5 | 2.31 | 43.3 | 2.49 | 1.36 | 53.0 | 2.39 | 0.00 | 0.00 | 0.85 | 0.14 |
| $Mg^{2+}$ | µeq/L | 0.82 | 11 | 12 | 5 | 1.43 | 22.3 | 2.97 | 0.83 | 18.3 | 2.57 | 0.00 | 0.00 | 0.09 | 0.00 |
| $HCO_3^-$ | µeq/L | 0.87 | 11 | 12 | 5 | 0.52 | 91.8 | -0.05 | 4.61 | 104 | 7.68 | 0.00 | 0.00 | 0.85 | 0.17 |
| Inorganic Nutrients | | | | | | | | | | | | | | | |
| TDP | P µg/L | 0.2 | 11 | 12 | 5 | 1.82 | 13.7 | 3.80 | 0.06 | 35.5 | 3.03 | 0.08 | 0.00 | 0.03 | 0.00 |
| SRP | P µg/L | 0.9 | 11 | 12 | 5 | 1.00 | 11.9 | 3.20 | 0.33 | 32.6 | 2.77 | 0.27 | 0.26 | 0.00 | 0.03 |
| TDN | N µg/L | 7 | 11 | 12 | 5 | 44.8 | 44.3 | 134 | 16.0 | 24.9 | 138 | 0.96 | 0.17 | 0.03 | 0.11 |
| $NO_2^- + NO_3^-$ | N µg/L | 2 | 11 | 12 | 5 | 11.9 | 9.08 | 6.00 | 5.99 | 10.7 | 3.67 | 0.00 | 0.00 | 0.06 | 0.36 |
| $NH_4^+$ | N µg/L | 3 | 11 | 12 | 5 | 24.6 | 23.6 | 90.2 | 8.81 | 17.4 | 110 | 0.86 | 0.04 | 0.03 | 0.04 |
| Organic Nutrients | | | | | | | | | | | | | | | |
| DOC | ppm | 0.06 | 11 | 12 | 5 | 0.15 | 0.49 | 0.40 | 0.06 | 0.59 | 0.25 | 0.12 | 0.61 | 0.00 | 0.68 |
| DOM C1 | FI | N/A | 10 | 9 | 5 | 3.24 | 3.72 | 3.22 | 2.94 | 3.71 | 2.24 | 0.76 | 0.50 | 0.99 | 0.63 |
| DOM C2 | FI | N/A | 10 | 9 | 5 | 5.27 | 6.40 | 3.28 | 4.27 | 4.41 | 1.25 | 0.58 | 0.91 | 0.33 | 0.03 |
| DOM C3 | FI | N/A | 10 | 9 | 5 | 1.63 | 6.44 | 21.2 | 1.46 | 6.48 | 28.5 | 0.04 | 0.00 | 0.00 | 0.00 |
| DOM C4 | FI | N/A | 10 | 9 | 5 | 2.96 | 4.69 | 2.74 | 2.39 | 3.49 | 0.94 | 0.22 | 0.28 | 0.85 | 0.09 |
| DOM C5 | FI | N/A | 10 | 9 | 5 | 1.95 | 4.78 | 6.77 | 2.25 | 5.48 | 5.29 | 0.15 | 0.02 | 0.03 | 0.03 |
| Microbial Assemblages | | | | | | | | | | | | | | | |
| Acidobacteria | % | N/A | 5 | 11 | 3 | 1.1 | 3.2 | 1.5 | 0.84 | 3.6 | 1.2 | 0.73 | 0.70 | 0.54 | 0.53 |
| Actinobacteria | % | N/A | 5 | 11 | 3 | 17 | 30 | 15 | 6.4 | 22 | 21 | 0.81 | 0.75 | 0.86 | 0.05 |
| Bacteroidetes | % | N/A | 5 | 11 | 3 | 14 | 9.2 | 16 | 5.2 | 15 | 18 | 0.00 | 0.03 | 0.82 | 0.04 |
| Chloroflexi | % | N/A | 5 | 11 | 3 | 0.7 | 8.1 | 6.1 | 0.51 | 5.6 | 9.1 | 0.01 | 0.00 | 0.20 | 0.00 |
| Cyanobacteria | % | N/A | 5 | 11 | 3 | 16 | 0.07 | 7.8 | 17 | 0.11 | 7.5 | 0.04 | 0.00 | 0.47 | 0.34 |
| Firmicutes | % | N/A | 5 | 11 | 3 | 1.0 | 10 | 0.06 | 1.9 | 15 | 0.10 | 0.10 | 0.24 | 0.54 | 0.22 |
| Gemmatimonadetes | % | N/A | 5 | 11 | 3 | 0.39 | 3.9 | 0.02 | 0.23 | 4.9 | 0.02 | 0.01 | 0.01 | 0.04 | 0.01 |
| Proteobacteria | % | N/A | 5 | 11 | 3 | 43 | 30 | 42 | 16 | 17 | 9.1 | 0.17 | 0.99 | 0.92 | 0.52 |

Technical corrections/comments:

1) Is the use of 'warm' and 'cold' in quotations necessary throughout? I feel it is not, as long as you state early on that these are the terms you are going to use.
Response: Quotations on 'warm' and 'cold' basal ice were removed, except for the last paragraph of the introduction when they are first introduced and defined.

2) The definition of cold based and warm based glaciers is repeated throughout the paper – it is only really necessary to define these terms once.
Response: Definitions were removed after the initial description of these terms

3) Consider revising the word 'parent' in 'parent ice' - this could lead to inference that the basal ice is always of younger age, which is not necessarily true. As such, this phrase may be slightly misleading. Consider revising throughout. If you choose to use parent ice – this should be defined and used consistently.
Response: Changed 'parent ice' to 'meteoric glacier ice' throughout.

4) There are many sentences which are poorly constructed, with use of multiple 'and/s', 'and/or' and 'also/s'. Often, this disrupts clarity and flow. Please consider revising.
Response: Several sentences were revised to remove 'and/or' (x8) and 'also' (x7). Lists were condensed where appropriate, for example: the last sentence of the first paragraph was changed to "Both basal ice and subglacial water are known to host populations of microbes that mediate redox reactions (e.g. Sharp et al., 1999; Wadham et al., 2004), play an active role in bedrock weathering (e.g. Tranter et al., 2002), and produce and/or consume ecologically important nutrients (e.g. Bottrell and Tranter, 2002; Boyd et al., 2011; Hodson, 2007; Statham et al., 2008; Tranter et al., 2002; Wadham et al., 2012)").

5) Line 32 - are you missing a reference related to subglacial microbial mediated chemical weathering?
Response: Tranter et al (2007) reference was added since the sentence is referring to weathering. Microbial mediated chemical weathering is discussed two sentences later and relevant references are included there.

6) Consider rephrasing line 39 for clarity – the first sentence is a little unclear, I think you may be missing a word.
Response: This sentence was rewritten: "Subglacial processes and the composition of basal ice can dramatically impact the biogeochemistry of meltwater and sediments exported from glaciers in a warming world. For example […]",

7) Line 44 274 - too many spaces.
Response: Removed space

8) Line 259, this would be an appropriate place to reference the Wadham (2016) study.
Response: The sentence was restructured to highlight and reference O'Donnell et al (2016) which we assume is the one you're referring to: "Excess $NH_4^+$ would be particularly prevalent during the

degradation of nitrogen-rich organic matter as has been identified in basal ice from other sites (O'Donnell et al., 2016), and observed in this study (protein-like DOM described by PARAFAC C1 and C2)"

9) Line 274 - consider rephrasing the sentence starting with 'Because: : :'.
Response: Sentence was rephrased and split into two: "The sedimentary rocks near/underlying the Western Margin support well-developed soils and vegetation. Therefore, even limited interaction with the substrate could have resulted in the acquisition of significant humic-like DOM in this cold-based system if this material was abundant in the substrate."

10) Line 289, although production and consumption of autochthonous OM are mentioned in Wadham (2016), I think there are other more appropriate references for this point.
Response: That should have been O'Donnell et al (2016) rather than Wadham et al (2016) so has been revised accordingly.

11 ) The font size of the figure captions vary.
Response: Original figure files can be provided for publishing to ensure consistent formatting.

12) Please, standardized units throughout. For example, currently, there is a mixing of DOC units mg L-1 and ppm.
Response: revised to consistently use ppm throughout

---

## Author Comment (AC3) · 27 Nov 2019

We would like to thank this anonymous reviewer for reviewing this manuscript – please find responses to each question below.

Line 64: I am not sure what the "a" refers too in "> 20 m a-1" but perhaps this is a common unit from studies on glaciers that I am unfamiliar with Response: a refers to annum and is a common unit for representing glacier velocity but has been changed to 'yr' for this biogeochemistry audience.

Line 199 - this seems like an odd way to report this. "less than half a percent of the OTUs" perhaps <0.5% would be clearer? Response: Sentence was revised to say "<0.5%"